# Carboxymethyl Cellulose/Gelatin Hydrogel Films Loaded with Zinc Oxide Nanoparticles for Sustainable Food Packaging Applications

**DOI:** 10.3390/polym14235201

**Published:** 2022-11-29

**Authors:** Aqsa Zafar, Muhammad Kaleem Khosa, Awal Noor, Sadaf Qayyum, Muhammad Jawwad Saif

**Affiliations:** 1Department of Chemistry, Government College University, Faisalabad 38000, Pakistan; 2Department of Basic Sciences, Preparatory Year Deanship, King Faisal University, Al-Hassa 31982, Saudi Arabia; 3Department of Applied Chemistry, Government College University, Faisalabad 38000, Pakistan

**Keywords:** hydrogel films, gelatin, antimicrobial activity, food packaging, TGA

## Abstract

The current research work presented the synthesis of carboxymethyl cellulose–gelatin (CMC/GEL) blend and CMC/GEL/ZnO-Nps hydrogel films which were characterized by FT-IR and XRD, and applied to antibacterial and antioxidant activities for food preservation as well as for biomedical applications. ZnO-Nps were incorporated into the carboxymethyl cellulose (CMC) and gelatin (GEL) film-forming solution by solution casting followed by sonication. Homogenous mixing of ZnO-Nps with CMC/GEL blend improved thermal stability, mechanical properties, and moisture content of the neat CMC/GEL films. Further, a significant improvement was observed in the antibacterial activity and antioxidant properties of CMC/GEL/ZnO films against two food pathogens, *Staphylococcus aureus* and *Escherichia coli.* Overall, CMC/GEL/ZnO films are eco-friendly and can be applied in sustainable food packaging materials.

## 1. Introduction

Today, food packaging is a growing technology because of rapid advancements in the fields of biopolymers and materials science. They are being used as new modes of food coatings. Encapsulation of biopolymers matrices, and food packaging materials for functional foods, proteins and polysaccharides have gained a lot of attention recently [1,2,3]. Active packaging is thought to be the best way to increase the safety and shelf life of food [4]. The presence of antimicrobial agents and antioxidants in active packaging plays a vital role in stopping biological or chemical changes and microbial growth in packaged foods [5]. In addition to antimicrobial agents, antioxidants, essential oils, natural pigments, and plant extracts have also been used in packaging materials [6]. Inorganic materials such as nano-sized metals, metal salts, and metal oxides seem promising for this purpose and are frequently used as antibacterial agents [7,8]. The incorporation of metal nanoparticles (ZnO, TiO_2_ and silica) into composite films has resulted in nanocomposites that are light in weight, stronger in thermomechanical performance, fire resistant, and less permeable to gases. They also reduce the flow of oxygen inside packaging containers when added to plastic films. To keep food fresh for a very long time, they serve as a barrier against gases and moisture [9]. The many properties of packaging materials are greatly enhanced by nanomaterials, which have a significant impact on the food packaging industry. One of the most significant categories of nanomaterials is zinc oxide nanoparticles to improve thermomechanical and antimicrobial properties [10]. Zinc oxide nanoparticles are now widely available commercially and are easy to be used directly due to their outstanding antibacterial properties, high thermal stability, excellent mechanical properties, and heat resistivity [11,12,13]. Tetrapod-shaped ZnO nanomaterials have recently been reported to inhibit herpes simplex virus (HSV) infection. It exhibits adjuvant-like properties, improves viral presentation to dendritic cells, and boosts humoral and cell-mediated immunity. ZnO-Nps have been evaluated for their antiviral properties against HSV-1, HSV-2, and influenza. This is primarily because ZnO nanoparticles have the ability to modulate the immune system [14,15]. As packaging materials and coating agents, natural polymers have numerous other uses. In the pharmaceutical sector, gelatin is commonly employed as an absorbent sheet, a wound dressing, an adhesive, and as an excipient in controlled drug delivery [16]. Several studies have shown that combining gelatin with polysaccharides, including alginate, chitosan, hyaluronic acid and other proteins, can enhance its properties due to their permeability and self-adhesiveness, as well as their capacity to form chemical and physical hydrogel films. They also serve as drug delivery and tissue regeneration support matrices. Cellulose is another biopolymer that is plentiful, renewable, and biodegradable and has been used to prepare biocompatible composite films [17]. Carboxymethyl cellulose (CMC), a cellulose derivative, has been extensively applied in cellular growth, food processing, food packaging, the pharmaceutical industry, and medical fields because of its good hydrophilicity, biocompatibility, and film formability [18,19]. The basic requirements for materials used in packaging are that they have good mechanical and thermal performance, protection against gases, and transparency. Therefore, the aim of the current research was to fabricate carboxymethyl cellulose and gelatin-based films using ZnO nanoparticles as a functional material. After characterization by FT-IR and XRD, CMC/GEL/ZnO composite films were investigated for the water vapor permeability (WVP), moisture contents, thermal and mechanical stability, antimicrobial activity, and free radical inhibition activity.

## 2. Materials and Methods

### 2.1. Chemicals

Gelatin powder (Food grade; Halal), carboxymethyl cellulose (CMC: pKa =3.5, with medium viscosity; Mw. 90,000), glutaraldehyde (GTA), sodium azide, zinc nitrate hexahydrate (Zn(NO_3_)_2_.6H_2_O), 2,2-diphenyl-1-picrylhydrazyl (DPPH), iron sulphate (FeSO_4_), 2,2-azino-bis-(3-ethylbenzothiazoline-6-sulfonic acid) (ABTS) and ferric chloride (FeCl_3_) were used as received. All chemicals were procured from Sigma-Aldrich Pakistan.

### 2.2. Chemical Characterization

For major functional groups and shifting of absorption bands in synthesized CMC/GEL/ZnO-nanocomposite films, IR spectrophotometer (Bruker) was used, and spectra were recorded in 4000–400 cm^−1^. TGA was determined, under nitrogen from room temperature to 700 °C, using Netzsch and Perkin Elmer TGA-7. Tensile strength and % elongation were determined by DMA Q800 V21.3 Build 96. Surface morphology was examined (FEI-NOVA Nano SEM-450, Hillsboro, OR, USA) by using Chroma meter (Konica Minolta, CR-400, Tokyo, Japan) with white standard colour plate (L_o_ = 92.15, a_o_ = −0.41, and b_o_ = 4.55) as a background, of surface colour (Hunter L, a, and b-values).

### 2.3. Green Synthesis of ZnO-Nps

Green chemistry is extensively used in research and is considered eco-friendly because it uses plant phytochemicals to prepare nanoparticles. Keeping this in mind, ZnO-Nps (<50 nm) were prepared using mint (Mentha longifolia) leaf extract. Fresh mint leaves were cleaned by washing with distilled water and were ground until very fine particle sizes were obtained, and were then air dried. About 10 g of fine powdered mint leaves in distilled water (250 mL) were heated at 70 °C. After cooling at room temperature for about 30–40 min, the heavy biomaterial that settled down after centrifugation at 4000 rpm was removed by filtration. The clear solution of mint extract was then stored at 4 °C and used for the preparation of nanoparticles. Following that, zinc oxide nanoparticles (50 nm) were prepared using mint leaves extract as an oxidizing agent as per the literature with slight modifications [20]. A mixture of 100 mL zinc nitrate Solution (0.01 M) and 30 mL of mint leave extract was heated at 60 °C with constant stirring until the bio-reduced ZnO nanoparticles settled down as white precipitates. The resulting, white-colored precipitates were dried at 80 °C for approximately six hours before being calcined at 600 °C for approximately two hours. 

### 2.4. Fabrication of Carboxymethyl Cellulose-Gelatin-ZnO Composite Films

To prepare carboxymethyl cellulose-gelatin-ZnO (CMC/GEL/ZnO) hydrogel films, the process of solution casting was employed for CMC/GEL (75:25). Briefly, CMC and GEL solutions (2% *w*/*v*) were mixed at 40 °C for 4 h along with 1 mL of glutaraldehyde as a crosslinking agent and 0.02% sodium azide to stop the bacterial growth. The CMC/GEL solutions were then slowly mixed with ZnO nanoparticles (1, 1.5, 2, 2.5 *w*/*w*) by sonication for two hours to obtain a clear solution. Then 10 mL of CMC/GEL/ZnO composite solution was poured into Teflon-made boats and dried at 25 °C. The films were peeled then off after drying in an oven at 70 °C for about twelve hours [21].

### 2.5. Moisture Contents and Water Vapor Permeability

To determine the moisture contents (MC), each CMC/GEL/ZnO nanocomposite film was cut into 2.5 × 2.5 cm square sizes and dried in an oven at 70 °C for about twelve hours. The difference in final and initial weight of film was noted as a moisture content. Whereas water vapor permeability (*WVP*) was determined gravimetrically. For this purpose, square-shaped pieces (6 × 6 cm) of the film were mounted on top of water vapor permeability measuring cups horizontally, having water (20 mL) and placed in the oven at 50 °C. The weight of the cups was noted at regular intervals of 30 min during twelve hours. [22]. After that *WVP* was calculated as:(1)$$WVP=\frac{\left(WVTR\times L\right)}{\Delta p}$$ Here, *WVTR*: rate of water vapor transmission (g/m^2^. s),

*L*: film thickness (m), Δ*p*: partial water vapor pressure differential across the film. 

### 2.6. Antimicrobial Assay

Six pathogens were tested using neat CMC, GEL, CMC/GEL, and CMC/GEL/ZnO nanocomposite films. Three of which were gram-positive: *Staphylococcus aureus* (ATCC 6538), *Bacillus subtilis* (ATCC 6633), and *Listeria monocytogenes* (ATCC 19111), and three of which were gram-negative: *Enterobacter aerogenase* (ATCC 13048), *Escherichia coli* (ATCC 15224), and *Bordetella bronchiseptica* (ATCC 4617). Briefly, bacterial strains were cultured for about 24 h in agar-agar nutrient broth at 37 °C. 1.5 mL of the broth culture of bacterial strains 10^4^–10^6^ (CFU/mL). In Petri dishes that had been sterilized, strains were added to an agar-agar medium and allowed to set at 45 °C. The solutions of CMC, GEL, CMC/GEL, and CMC/GEL/ZnO nanocomposite films in DMSO (10 mg/mL) were then added to each well. Each bacterial strain was prepared in triplicate and was incubated for about 24 h at 37 °C. By measuring the size of the inhibition zone, antibacterial activity was calculated (mm). Cefixime (1 mg/mL), a standard antibiotic, was used as a positive control [23,24].

### 2.7. Antioxidant Assays

The DPPH method was used to measure activity that scavenges free radicals of neat CMC, neat GEL, CMC/GEL, and CMC/GEL/ZnO nanocomposite films [25]. Stock solutions (5 mg/mL) of neat CMC, GEL, and nanocomposite films were made in DMSO. Serial dilutions of 5, 10, 20, 40, 100, and 200 g/mL were performed. In glass vials, 15 µL of each film solution and solution of DPPH (0.1 mM) were mixed together and diluted to 3 mL with methanol. For about 45 min, the reaction mixture was incubated at 37 °C in a dark chamber; this caused the DPPH solution’s color to change from deep violet to light yellow. At 517 nm, absorbance was noted by using a spectrophotometer. A standard, butylated hydroxyanisole, was used for each experiment, which was carried out in triplicate (BHA, 5 mg/mL). The decreased absorbance of the mixture indicated higher radical inhibition activity. The % age of scavenging activity was calculated as:$$\%\;\mathrm{scavenging}\;\mathrm{activity}=\frac{\mathrm{absorbance}\;\mathrm{of}\;\mathrm{control}-\;\mathrm{absorbance}\;\mathrm{of}\;\mathrm{test}\;\mathrm{sample}\;}{\mathrm{absorbance}\;\mathrm{of}\;\mathrm{control}}\times 100$$

## 3. Statistical Analysis

The obtained data were verified statistically by applying ANOVA using Minitab Software. Each experiment was carried out in triplicate. The statistical significance value was tested at *p* < 0.05. 

## 4. Results and Discussion

### 4.1. IR Investigation

The FT-IR absorption spectra of neat carboxymethyl cellulose, gelatin, CMC/GEL/glutaraldehyde, and CMC/GEL/ZnO nanocomposites are shown in Figure 1. Peak positions and modes of interaction are presented in Table 1. The crosslinking between CMC/GEL and ZnO-Nps was responsible for the majority of the changes in absorption frequencies that were seen. The bands at 2910, 1600, and 1440 cm^−1^ in CMC are due to stretching vibrations of aliphatic C-H, (-COO)_asy._ and (-COO)_sym_, respectively (Figure 1a). The absorption bands of glucosidic units (C-O-C) appeared at 1160 and 1070 cm^−1^. A broad absorption band at 3390 cm^−1^ was attributed to the hydrogen bonding of OH groups of absorbed water and secondary alcohols (CMC) [26,27]. Absorption bands at 3390 and 3315 cm^−1^ correspond to the hydroxyl and amino groups of gelatin film, respectively (Figure 1b). The absorption bands at 1649 and 1551 cm^−1^ are due to the (C=O) and (C-N) absorption bands of amide I and amide II in the gelatin structure. The interaction of CMC’s anionic groups with thegelatin’s cationic groups was confirmed by the shifting of peaks from 3315, 1450 to 3328, 1460 cm^−1^ (Figure 1c). The absorption band of the C-H group has been shifted from 2910 to 2955 cm^−1^ in CMC/Gel/GTA (Figure 1d). The reaction of GTA with CMC/GEL was confirmed by an absorption band below 1210 cm^−1^ forming a hemiacetal structure. Similarly, homogenous mixing of ZnO-Nps with CMC/GEL film (Figure 1e) was confirmed by an absorption peak at 470 cm^−1^ [28].

### 4.2. XRD Analysis

The homogenous mixing of ZnO nanoparticles in the CMC/GEL film was confirmed by the X-ray diffraction pattern, which also quantified the changes from an amorphous to crystalline structure. The XRD pattern of ZnO-Nps (a), CMC/Gel/ZnO (b), CMC/Gelatin (c), CMC (d), and Gelatin (e) is shown in Figure 2. The sharp peaks of the ZnO nanoparticle (Figure 2a) that correspond with the data in JCPDS No. 36-1451, confirmed the wurtzite crystal structure with a hexagonal phase [29]. The diffraction pattern of CMC/GEL/ZnO composite film revealed a characteristic peak of ZnO at 2θ; 32.130, 34.620, 36.460, 46.930, 53.740, 56.460, and 61.350 with low intensity, which indicated the homogenous mixing of ZnO-Nps in CMC/GEL film (Figure 2b). The CMC has an amorphous structure, while gelatin is partially crystalline with peaks at 2θ = 7.6° and 22.31°, Figure 2d,e. In Figure 2c–e, there is no sharp diffraction peak, and an increased diameter of the interreticular triple helix with increased intensity is observed due to the mixing of CMC and gelatin. The interaction between CMC and gelatin has been confirmed by the absence of X-peaks. This phenomenon shows the reduction of hydrogen bonding between the hydroxyls of gelatin and the cellulose groups of CMC. The cross-linking between CMC and gelatin films by GTA exhibited a negligible triple helix approaching 0.8% are in close agreement with the literature [30].

### 4.3. Morphological Studies

All the prepared CMC/gelatin/ZnO nanocomposites were mechanically flexible and self-standing. SEM was used to examine the surface morphology of CMC, gelatin, and ZnO-nanocomposite films and images are shown in Figure 3. The interaction of zinc oxide nanoparticles with CMC and gelatin controls the networked microstructure. Without ZnO nanoparticles, neat CMC, gelatin biopolymer, and CMC/gelatin blend films showed a uniform surface, demonstrating that the film-forming polymers were mostly amorphous and had little crystallization. While the CMC/gelatin film with zinc oxide-Nps additions had a heterogeneous rough surface, with the zinc nanoparticles being uniformly distributed throughout the films and preventing particle separation. It explained that how metal nanoparticles prevented particle aggregation and created high-viscosity CMC/gelatin films, thus a highly stable CMC/gelatin mixture was used [31].

### 4.4. Optical Properties

Optical properties of CMC/GEL-based films loaded with various weight ratios (2.5% *w*/*w*) of ZnO-Nps were studied, and data is shown in Table 2 and in Figure 4. Neat gelatin film had two peaks for typical CMC and gelatin peaks at 230–240 nm and 265–280 nm. The nanocomposite films containing ZnO nanoparticles, on the other hand, revealed an additional peak for the ZnO-nps near 370 nm. The peak intensity increased with increasing ZnO nanoparticle contents in CMC/gelatin matrices from 1% to 2.5% [32].

The L-values of the CMC, gelatin, and CMC/gelatin films exceeded 91, and the a and b parameters ranged from 0.8 to 0.3 and 5.3 to 6.7, respectively, making the films highly transparent. By incorporating ZnO nanoparticles, L and a values were significantly shifted to the lower end, while b values increased. As a result of these changes, the total colour difference between the films increased significantly (Δ*E*) [33]. The Δ*E* was calculated by:$$\Delta E={\left({\left(\Delta L\right)}^{2}+{\left(\Delta a\right)}^{2}+{\left(\Delta b\right)}^{2}\right)}^{0.5}$$
where *ΔL, Δa*, and *Δb* are the differences between value of standard colour plate and composites films. In the presence of UV rays, food ingredients oxidise, destroying nutrients and bioactive compounds. Film transparency and UV protection are essential optical properties for applications in food packaging. For the sake of food safety and quality, oxidation reactions that result in toxic substances, off flavours, discoloration, or rancidity are not allowed [34].

Furthermore, the % transmittance of light was measured at *T_280_* and *T_660_* nm. At *T_660_* nm and *T_280_* nm, the neat CMC and gelatin films had transmittance values of 87.2 ± 0.50, 90.7 ± 0.15, 58.3 ± 1.25, and 30.4 ± 1.50, respectively. When ZnO-Nps was added to CMC/gelatin blend, the percentage transmittance of films at 280 nm was drastically reduced, falling to just 0.3%, whereas the percentage transmittance values at 660 nm only slightly decreased as ZnO-Nps concentrations were increased [35]. The amount of ZnO-Nps present had a significant impact on the transmittance value of the nanocomposite films.

### 4.5. Thermal Stability

The TGA technique was used to study the thermal properties of pure biopolymer films as well as films containing ZnO-Nps, and thermograms are shown in Figure 5, which demonstrate the degradation patterns of neat CMC/GEL films and CMC/GEL/ZnO-Nps. All films showed multiple steps of thermal degradation. The first step of degradation started around 95 to 100 °C, with a percentage weight loss ranging from 10 to 20% because of the evaporation of loosely bound gases from films. The second stage of degradation of amino groups in CMC/GEL film was observed at 290–360 °C with 84% weight loss. The other loss in weight was observed at 415 and 75 °C, which were considered the third and fourth steps of degradation in weight, respectively [33]. Compared to neat CMC/GEL films, the degradation curves of prepared nanocomposites films are shifted towards higher temperatures because biopolymers and ZnO-Nps have a strong interaction and are thoroughly mixed [36].

### 4.6. Mechanical Properties

Tensile strength, percent elongation at break (EB), and elastic modulus, which are key factors in determining the strength and flexibility of the film, were studied as mechanical properties. The effect of ZnO-Nps on EB, tensile strength, and elastic modulus of CMC/GEL films was determined, and the data is presented in Table 3. The addition of ZnO-Nps to the CMC/GEL films has significantly improved the flexibility (EB) and strengthened the CMC/GEL film. Initially, the CMC/GEL (75:25) film had tensile strength and elongation at break values of 39.25 MPa and 4.41 0.23%, respectively. The incorporation of ZnO nanoparticles has increased the elongation at break, and tensile strength values for the CMC/GEL films to 44.6 MPa and 10%, respectively. The increased tensile strength of CMC/gelatin films may be attributed to the incorporation of the proper amount of nanoparticles as well as the bonding between hdroxy groups of gelatin and the hydrophilic groups of CMC, resulting in the formation of a mechanically stable nanocomposite between CMC, gelatin, and zinc oxide nanoparticles [35].

### 4.7. Moisture Content and Water Vapor Permeability 

The moisture contents (MC) and water vapour permeability (WVP) of neat CMC, neat GEL, CMC/GEL, and MCM/GE/ZnO- films were determined and the data are presented in Table 1. CMC/GEL films had slightly lower moisture contents and water vapour permeability, whereas CMC/GEL/ZnO-Nps films had higher values and these values increased with higher zinc oxide nanoparticle concentration. ZnO nanoparticles in the CMC/GEL film matrix make the films porous, increasing water vapour permeability.

### 4.8. Antibacterial Results

The antimicrobial screening data of CMC, GEL, and ZnO films are presented in Table 4. As per the literature and experiments, the CMC/GEL film had no antibacterial action against both types of tested microorganisms; however, the composite films incorporating ZnO-Nps had marked antibacterial activity. The antimicrobial activity of composites depends on the nature and type of inorganic fillers and the type of bacterial strains used. On both types of tested bacterial strains, the CMC/GEL hydrogel films had a bacteriostatic effect due to the presence of ZnO-Nps, which delayed the growth of the germs because the structure of their cell walls was different. Generally, gram-negative bacteria have a complex cell wall made of a thin layer of peptidoglycan protected by an extra outer membrane, in contrast to gram-positive pathogens that have a thick cell wall with many layers of peptidoglycan. The size and shape of zinc oxide nanoparticles in CMC/GEL/ZnO composite films mainly increased the antibacterial activity of tested bacterial strains. Though the antibacterial mechanism of ZnO is unknown, it is thought that zinc oxide nanoparticles increased the permeability of the membrane around the bacteria, allowing metal nanoparticles to readily pass through the bacterium’s cell wall. Various reactive oxygen species (ROS) such as hydroxyl radicals (OH^−^), hydrogen peroxide (H_2_O_2_), superoxide anions (O^2−^), and organic hydroperoxides are produced inside the cell during this process. When ROS overcomes the cellular antioxidant defense mechanism, oxidative stress occurs, which is linked to damage of several critical macromolecules that ultimately leads to cell death [34]. 

### 4.9. Antioxidant Activity

The antioxidant activity of neat CMC, GEL, CMC/GEL and CMC/GEL/ZnO nanocomposite films was evaluated by their DPPH (1,1-diphenyl-2-picrylhydrazyl) free radical scavenging activity. The effect of tested nanocomposite materials on generation of free radical of DPPH or radical scavenging activity is measured at 517 nm by decrease in molar absorptivity of DPPH. The degree of discoloration reveals the antioxidant compounds’ scavenging capacity in terms of H-donating capacity. Furthermore, the ability of antioxidant compounds to scavenge free radicals depends on their concentration. The radical-scavenging activity rises with antioxidant compound concentration, and a low *IC_50_* value (i.e., 50% inhibitory concentration) indicates a high antioxidant activity. The obtained results are shown in Table 5. Synthesized composite films containing ZnO-Nps showed free radical scavenging activity with IC_50_ in the range of 39 to 85 µg/mL while their respective matrix polymers (CMC/GEL) showed activity more than 100 µg/mL. On comparison of IC_50_ values of CMC, GEL and their composite films with different wt ratio of ZnO-Nps, it was observed that such CMC/GEL/ZnO-nanocomposites are more active for DPPH activity than their parent polymers [35,36].

## 5. Conclusions

Owing to the versatile nature of biopolymer composites, ZnO nanoparticles have drawn a lot of attention in the field of food packaging. The purpose of this research was to synthesize zinc oxide nanoparticles (<50 nm) by using plant extract as a reducing agent, and to incorporate this in CMC/gelatin matrices to assess the optical, thermal, mechanical and microbial properties. ZnO-based multifunctional CMC/GEL nanocomposites were prepared. ZnO-Nps affected the bio-functional and physical properties of CMC/gelatin composite films. The FTIR and XRD results confirmed the uniform dispersion of ZnO-Nps in the CMC/GEL, matrix to prepare compatible films. The addition of ZnO-Nps (1–2.5 wt. %) to CMC/GEL improved the thermal stability, mechanical properties, moisture contents, and water vapor permeability. CMC/GEL/ZnO-2.5% composite film showed good thermal stability and mechanical properties. CMC/GEL/ZnO 2.5% composite film showed strong antibacterial activity against foodborne pathogenic bacteria, *E. coli*, and *L. monocytogenes* and had high antioxidant activity. CMC-based composite films, such as CMC/GEL/ZnO 2.5%, showed improved thermo-mechanical properties with higher antioxidant and antibacterial activity as compared to CMC/GEL blend films. On the basis of the obtained results, CMC/GEL/ZnO nanocomposite films can be used to prevent photooxidation, ensure food safety, and increase the shelf life of packaged goods in active food packaging applications.

## Figures and Tables

**Figure 1 polymers-14-05201-f001:**
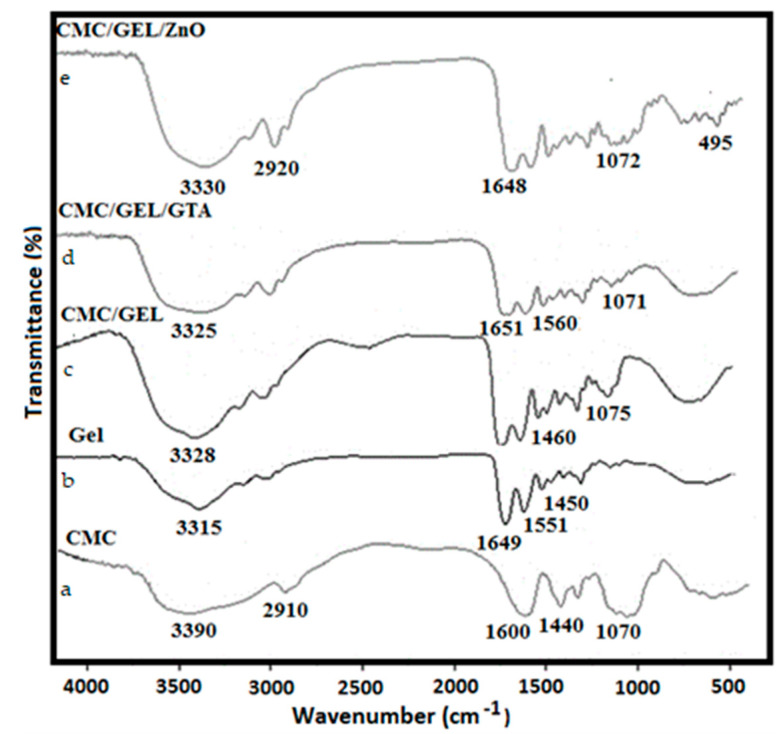
FT-IR Spectra of CMC/Gel/ZnO-Nps hydrogel films: (**a**) CMC, (**b**) Gel, (**c**) CMC/GEL, (**d**) CMC/GEL/GTA, (**e**) CMC/GEL/ZnO.

**Figure 2 polymers-14-05201-f002:**
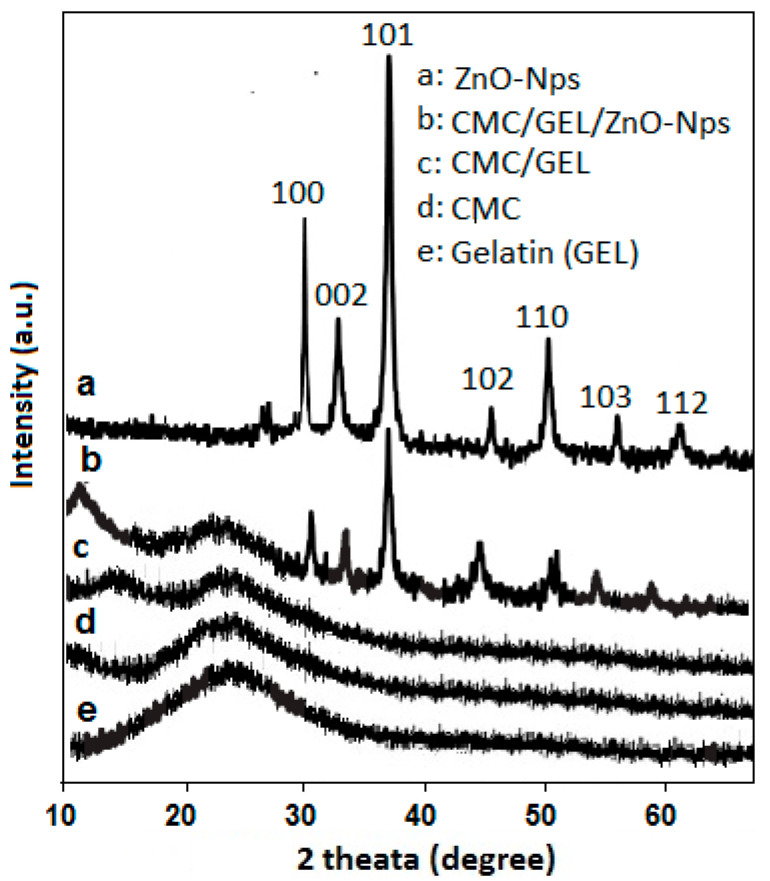
XRD pattern of CMC/Gel/ZnO nanocomposites hydrogels.

**Figure 3 polymers-14-05201-f003:**
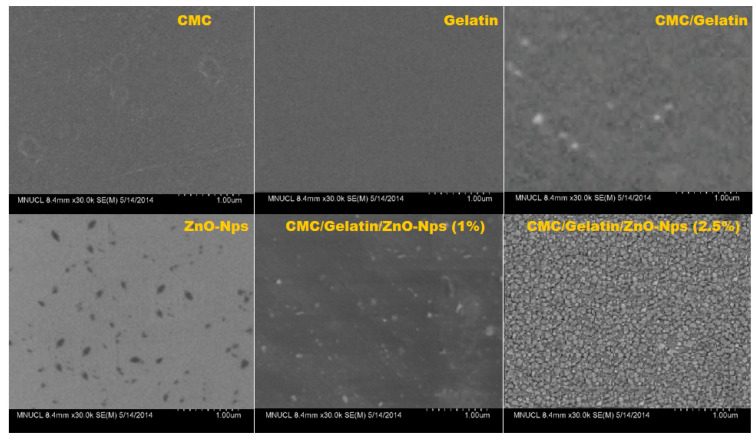
Scanning microscopy images of neat CMC, gelatin, CMC/gletan and CMC/gelatin/ZnO-Nps.

**Figure 4 polymers-14-05201-f004:**
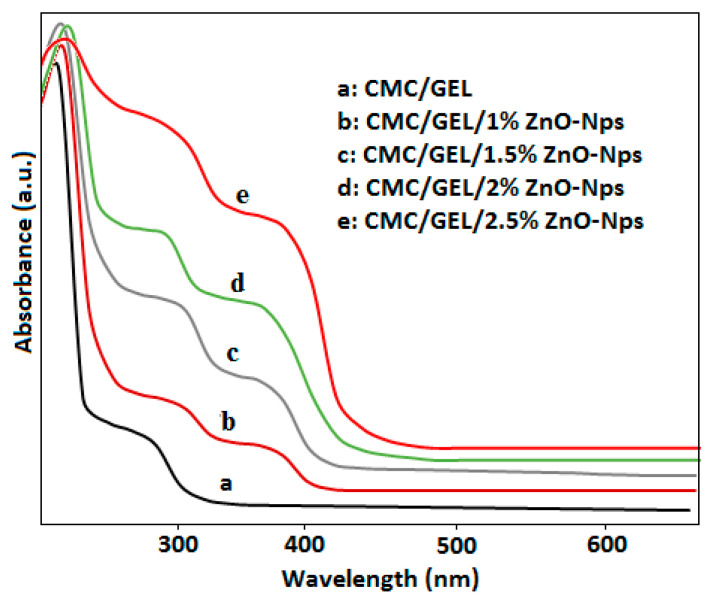
Optical studies of CMC/gelatin/ZnO-Nps.

**Figure 5 polymers-14-05201-f005:**
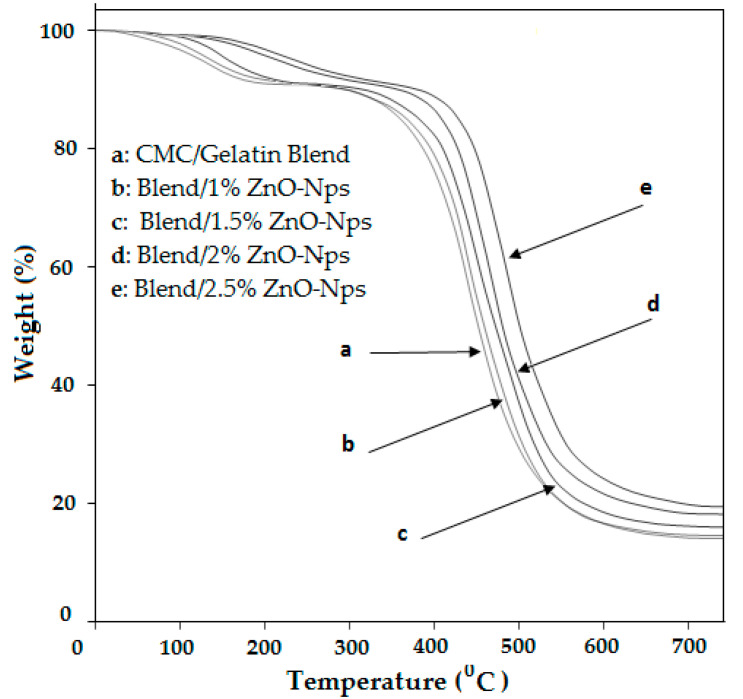
TGA of CMC/Gel/ZnO-nanocomposite hydrogels.

**Table 1 polymers-14-05201-t001:** Peak positions and vibration modes in CMC/GEL/ZnO nanocomposites hydrogels.

Wavenumber (cm^−1^)	Assignment	Ref.	Wavenumber (cm^−1^)	Assignment	Ref.
3350–3390	Stretching vibration of -OH group	[22]	1400–1460	(COO)_sym_	[24]
2910–2920	Asym. -CH_2_ stretching	[23,24]	1160–1165	(C-O-C) Ether stretching	[24]
1600–1649	(COO)_asy_	[22,24]	1070–1075	Stretching CH-O-CH_2_	[24,25]
1551–1560	C-N group	[23,24]	839–860	Asymmetric rocking	[26]

**Table 2 polymers-14-05201-t002:** Optical data of CMC/gelatin/ZnO composite films loaded with different contents of ZnO-Nps.

Sample	L	a	b	Δ*E*	*T_280nm_* (%)	*T_660nm_* (%)
Neat CMC	91.45 ± 0.15	−0.30 ± 0.15	5.3 ± 0.1	1.12	58.3 ± 1.25	87.2 ± 0.50
Neat Gelatin	91.31 ± 0.15	−0.50 ± 0.25	6.2 ± 0.50	1.89	30.4 ± 1.50	90.7 ± 0.15
CMC/Gelatin	91.20 ± 0.15	−0.80 ± 0.15	6.7 ± 0.15	2.40	25.6 ± 0.25	88.4 ± 1.25
CMC/Gelatin/ZnO^1^	90.44 ± 0.25	−1.24 ± 0.15	6.9 ± 0.50	3.06	14.5 ± 1.50	85.3 ± 0.25
CMC/Gelatin/ZnO^1.5^	90.27 ± 1.50	−1.31 ± 0.50	7.3 ± 0.25	3.26	2.4 ± 0.50	81.7± 1.15
CMC/Gelatin/ZnO^2^	90.64 ± 1.25	−1.38 ± 0.50	7.1 ± 0.15	3.15	1.5 ± 0.15	72.5± 1.50
CMC/Gelatin/ZnO^2.5^	90.85 ± 1.50	−1.46 ± 0.25	7.6 ± 0.15	3.50	0.9 ± 0.25	68. 5 ± 0.15

± standard deviation. (*p* < 0.05).

**Table 3 polymers-14-05201-t003:** Mechanical properties and Physical Data of CMC/GEL/ZnO nanocomposite films.

Components	Tensile Strength ± SD, (MPa)	Young’s Modulus ± SD, (MPa)	Elongation at Break ± SD, (%)	Moister Contents	Water Vapor Permeability (g m^−1^ day^−1^ atm^−1^)
Neat CMC	35.15 ± 1.25	1186 ± 0.06	14.18 ± 0.06	12.5 ± 1.15	1.15 ± 0.15
Neat GEL	37.25 ± 1.5	1390 ± 2.5	2.34 ± 0.06	10.25 ± 0.5	1.25 ± 1.5
75 CMC: 25 GEL Blend	39.25 ± 2.0	1255 ± 5.6	4.41 ± 0.23	11.5 ± 0.5	2.15 ± 0.5
Blend: 1 ZnO	40.15 ± 6.4	1050 ± 2.8	6.5 ±2.5	13.5 ± 1.5	2.55 ± 0.05
Blend: 1.5 ZnO	41.9 ± 1.25	970 ± 1.86	8.2 ± 1.5	13.8 ± 0.5	2.84 ± 1.25
Blend: 2 ZnO	42.5 ± 1.4	895 ± 7.25	8.58 ± 2.25	14.1 ± 0.0	2.92 ± 1.5
Blend: 2.5 ZnO	44.6 ± 2.5	765 ± 9.5	10.0 ± 1.25	14.5 ± 0.5	3.11 ± 0.05

**Table 4 polymers-14-05201-t004:** Antibacterial activity of CMC/GEL/ZnO-Nps hydrogel films.

Compound No.	Zone of Inhibition (mm) + St. Dev.					
	*Staphylococcus aureus*	*Bacillus subtilis*	*Listeria monocytogenes*	*Enterobacter aerogenase*	*Escherichia coli*	*Bordetella bronchiseptica*
Neat CMC	-	-	-	-	-	-
Neat GEL	-	-	-	-	-	-
75 CMC: 25 GEL Blend	12 ± 1.0	14 ± 1.25	-	-	13 ± 1.15	-
Blend: 1% ZnO	18 ± 1.05	22 ± 1.5	20 ± 1.15	15 ± 2.0	17 ± 1.15	14 ± 1.25
Blend: 1.5% ZnO	22 ± 1.0	25 ± 1.25	23 ± 1.5	19 ± 1.75	20 ± 1.5	19 ± 1.5
Blend: 2% ZnO	28 ± 2.0	26 ± 1.5	27 ± 1.0	22 ± 1.5	24 ± 1.5	21 ± 1.25
Blend: 2.5% ZnO	30 ± 1.03	29 ± 2.0	32 ± 1.0	26 ± 1.25	31 ± 1.25	24 ± 2.0
Cefixime	33 ± 1.5	31 ± 1.0	35 ± 1	29 ± 0.5	36 ± 1	31 ± 2

± = SD, standard deviation, (*p* ˂ 0.05 vs. control); 5–10 mm zone of inhibition (Activity present): 11–25 mm zone of inhibition (Moderate activity); 26–40 mm zone of inhibition (Strong activity).

**Table 5 polymers-14-05201-t005:** Antioxidant results of CMC/GEL films loaded with different concentration of ZnO-Nps.

Compound	% Scavenging ± sd Concentration µg / mL						IC_50_ µg/mL
	200	100	40	20	10	5	
Neat CMC	65 ± 1	58 ± 1	47 ± 1	38± 2	29± 1	15 ± 1	>200
Neat GEL	61 ± 1	56 ± 1	49 ± 1	33 ± 1	26 ±1	18 ± 1	>150
75 CMC: 25 GEL Blend	68 ± 1	59 ± 2	52 ± 2	44 ± 2	38 ± 1	22 ± 1	>100 ± 1
Blend: 1 ZnO	71 ± 1	66 ± 2	59 ± 1	38 ± 1	29 ± 1	24 ± 1	85 ± 1
Blend: 1.5 ZnO	74 ± 2	65 ± 1	54 ± 1	42 ± 1	35 ± 2	20 ± 1	62 ± 1
Blend: 2 ZnO	79 ± 1	66 ± 1	58 ± 1	43 ± 2	33 ± 1	18 ± 1	54 ± 1
Blend: 2.5 ZnO	84 ± 1	69 ± 1	65 ± 2	52 ± 1	48 ± 1	22 ± 1	42 ± 1
Butylated hydroxyanisole (BHA)	89 ± 0.5	85 ± 1	76 ± 0.25	68 ± 0.5	45 ± 1	15 ± 1	8 ± 0.05

± = SD, standard deviation, (*p* ˂ 0.05 vs. control); Antioxidant activity: IC_50_ 0–10 mgmL^−1^, very strongly active; 10–50 mgmL^−1^, strongly active; 50–100 mgmL^−1^, moderately active; 100–250 mgmL^−1^, weakly active; >250 mgmL^−1^, inactive.

## Data Availability

The authors declare that data supporting the findings of this study are available within the article entitled: “Carboxymethyl cellulose/gelatin hydrogel films loaded with Zinc Oxide nanoparticles for sustainable food packaging applications”.

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
