# Peer review of "Carboxymethyl Cellulose/Gelatin Hydrogel Films Loaded with Zinc Oxide Nanoparticles for Sustainable Food Packaging Applications"

_polymers, 2022, doi:10.3390/polym14235201_

Round 1
Reviewer 1 Report
The paper presents the synthesis and characterization of CMC/GEL/ZnO-Nps hydrogel films. The paper can be accepted for publication after major revision. The following issues should be clarified:
Information on the advantages and disadvantages of the obtained nanocomposite in comparison to existing analogs should be provided.
The discussion about the potential virucidal properties of the ZnO-Nps-based nanocomposites should be added in the introduction to interest the Readers.
Please write "carboxymethylcellulose" in the same manner. In one case, it's carboxymethylcellulose, and in another carboxymethyl cellulose.
What type of carboxymethylcellulose was used? Please suggest characteristics (producer, trademark, molecular weight, viscosity).
What type of gelatin was used?
The information about ZnO-Nps (size, shape, and charge) is absent in the paper.
Please add a scheme to illustrate the fabrication of carboxymethylcellulose-gelatin-ZnO composite films.
Please provide information from Fig.1 in the form of the Table where the wavenumbers of the peaks with assigned groups are marked.
The differential thermal analysis would be very useful to compare the impact of the ZnO-Nps on the thermal properties of the nanocomposites.
Please broaden the discussion about the impact of the ZnO-Nps on the mechanical properties of the nanocomposite.
I suggest citing similar work where similar results were intensively discussed.
https://doi.org/10.1016/j.cej.2022.137048
Author Response
The response to reviewers comments is attached for your consideration

Reviewer 2 Report
The study by Zafar et al., is a timely research topic. Biopolymers for active food packaging is gaining more attention due to side effects caused by conventional packaging materials. However, the experimental data is not enough and should be extended for broad applications of CMC/GLE/ZnO NP films.
1. Add more explanation about antimicrobial biofilms and their application in food sector, check these articles for appropriate literature review in introduction section.
· Silver Decorated Bacterial Cellulose Nanocomposites as Antimicrobial Food Packaging Materials
· Development and characterization of plant oil-incorporated carboxymethyl cellulose/bacterial cellulose/glycerol-based antimicrobial edible films for food packaging applications
2. Provide Reference for no antibacterial activity of CMC/GLE films at line 208.
3. Provide images for antibacterial activity of films to verify data provided in Table 2.
4. Change the heading of section 4.1 to Chemical characterization
5. Change “Carboxymethylcellulose” to CMC at line 137, also check whole manuscript for abbreviations.
6. Why authors did not carry further experiments with ‘CMC/Gel/GTA’ except FTIR and XRD?
7. The XRD should be repeated there is too much background noise.
8. For food packaging applications biocompatibility of films is an important property, as packaging may cause food toxicity.
9. For food packaging and biocompatibility evaluation refer to article “Development and Characterization of Yeast-Incorporated Antimicrobial Cellulose Biofilms for Edible Food Packaging Application”
10. Italicize the name of bacterial species in the abstract and throughout the whole manuscript.
11. Crosscheck the language of manuscript, several typos and grammatical errors were noticed.
Author Response

(The authors gave the same response as above.)

Reviewer 3 Report
Although the paper is interesting, it should be improved according to following lines:
+ Authors should discuss the amount of ash content on each sample regarding TGA test.
+ Authors did not discuss the antimicrobial properties of other materials in comparison with ZnO used in this study.
+ Relation to previous discussion, some new antimicrobial materials that can be discussed in comparison with ZnO are calcium iodate and silver. A publication regarding this: Gel diffusion-inspired biomimetic calcium iodate/gelatin composite particles: Structural characterization and antibacterial activity, Journal of Solid State Chemistry 285, 121262 (2020).
+ The thickness of produced films should be measured and discussed.
+ SEM, EDX and elemental mapping should be measured to indicate the surface properties of films along with the elemental properties.
+ The optical properties of samples should be measured by reflectance spectroscopy.
+ The images of samples and their haziness before and after inclusion of particles should be illustrated.
Author Response
Response to reviewers comments is attached for your consideration

Round 2
Reviewer 1 Report
After revision, the quality of the paper was essentially improved. I can recommend the paper for publication after minor revision. All text must be carefully checked, some mistakes are observed. For example, reference 13 should be carefully checked and corrected. There are names instead of surnames. The page number is not separated by a comma from the volume
Now is "Ostap L.; Yana, S.; Yurij, S.; Joanna, R.; Andre, G. S.; Taras, P.; Andrzej, B. Passive antifouling and active self-disinfecting antiviral surfaces. Rev. Chem. Eng. J. 2022, 446137048."
it will be correctly "Lishchynskyi O.; Shymborska, Y.; Stetsyshyn, Y.; Raczkowska, Y.; Skirtach, A. G.; Peretiatko, T.; Budkowski, A. Passive antifouling and active self-disinfecting antiviral surfaces. Rev. Chem. Eng. J. 2022, 446, 137048."
Author Response
Dear Reviewer
Your valuable suggestions are welcomed and addressed properly.

Reviewer 2 Report
Accept
Author Response
Reviewer 2 has not suggested any corrections in round 2
Reviewer 3 Report
It is acceptable now.
Author Response
Reviewer 3 has not suggested further corrections in round 2